# The bHLH Transcription Factors in Neural Development and Therapeutic Applications for Neurodegenerative Diseases

**DOI:** 10.3390/ijms232213936

**Published:** 2022-11-11

**Authors:** Dong Gi Lee, Young-Kwang Kim, Kwang-Hyun Baek

**Affiliations:** 1Joint Section of Science in Environmental Technology, Food Technology, and Molecular Biotechnology, Ghent University, Incheon 21569, Korea; 2Department of Biomedical Science, CHA Stem Cell Institute, CHA University, Seongnam 13488, Korea

**Keywords:** AtN conversion, basic helix-loop-helix, cell therapy, gene therapy, iPSC, reprogramming, TRANsCre-DIONE, transdifferentiation, tunneling nanotubes

## Abstract

The development of functional neural circuits in the central nervous system (CNS) requires the production of sufficient numbers of various types of neurons and glial cells, such as astrocytes and oligodendrocytes, at the appropriate periods and regions. Hence, severe neuronal loss of the circuits can cause neurodegenerative diseases such as Huntington’s disease (HD), Parkinson’s disease (PD), Alzheimer’s disease (AD), and Amyotrophic Lateral Sclerosis (ALS). Treatment of such neurodegenerative diseases caused by neuronal loss includes some strategies of cell therapy employing stem cells (such as neural progenitor cells (NPCs)) and gene therapy through cell fate conversion. In this report, we review how bHLH acts as a regulator in neuronal differentiation, reprogramming, and cell fate determination. Moreover, several different researchers are conducting studies to determine the importance of bHLH factors to direct neuronal and glial cell fate specification and differentiation. Therefore, we also investigated the limitations and future directions of conversion or transdifferentiation using bHLH factors.

## 1. Introduction

Transcriptional regulation is essential for the control of gene expression to correct the function of cells during development and throughout life. The basic Helix-loop-Helix (bHLH) superfamily, one of the transcription factors, has critical roles in tissue development and maintenance [1]. During the development of the nervous system, proneural bHLH transcription factors have critical roles in the regulation of cell proliferation, neuronal differentiation, and specification [2,3,4].

The helix-loop-helix (HLH) domain is situated at the N-terminus and is comprised of 40 to 50 basic amino acid residues [5,6]. Transacting factors are the proteins that bind to the promoter region of genes (*cis* elements). At the transcriptional level, they control certain physiological or biochemical processes. A DNA-binding domain, a transcriptional regulatory domain, a nuclear localization domain, and an oligomerization site are all found in the protein structure of transcription factors. In addition, an activation domain and an inhibitory domain might also be found in the transcriptional regulatory domain [7].

The polymorphism of bHLH transcription factors is weakly associated with neuropsychiatric, neurodevelopmental disorders, and neurodegenerative diseases. In neuropsychiatric disorders, for example, polymorphisms within the *Neurog1* regulatory region are associated with language deficits, and *Chr 5q31* containing the *Neurog1* locus is also associated with schizophrenia [8]. Among patients with schizophrenia, patients with *Neurog1* polymorphism showed decreased verbal memory, language ability, and visuospatial abilities [8]. Although *Neurog1* polymorphism is not the cause of schizophrenia, this polymorphism is more severe in patients with schizophrenia [9]. Mutations in *Neurod2* are associated with schizophrenia and schizoaffective disorder, and *Neurod2* polymorphism in patients with schizophrenia is associated with decreased verbal memory and executive functions [9].

In neurodegenerative diseases, late-onset neurodegenerative diseases appear to be less associated with *bHLH* mutations, and only the *Ascl1* gene is associated with neurodegenerative diseases. *Ascl1* polymorphism is associated with Parkinson’s disease. In Parkinson’s disease, *Ascl1* is known to affect the developmental control of the coeruleus, a region thought to have neuroprotective effects through interaction with *Phox2b* [10]. *Neurog2* was also investigated for a potential association with Parkinson’s disease but was found to be of low association [11]. Finally, *Neurod6* is downregulated in Alzheimer’s disease and has been proposed as a potential biomarker [12].

The field of induced pluripotent stem cells (iPSCs) has created many gateways for research in therapeutics. They possess unique properties of self-renewal and differentiation to many types of cell lineage. Thus, it could replace the use of embryonic stem cells (ESC) and may overcome the various ethical issues regarding the use of embryos in research and clinics. iPSCs play a critical role in studying the molecular mechanisms of many diseases. Researchers are currently conducting different studies in differentiating various cell types from iPSCs, such as NPCs, by using bHLH transcription factors in vivo and in vitro.

Neural progenitor cells (NPCs) are responsible for producing various types of glial cells and neurons that make up the nervous system during brain development [13,14,15]. However, some of them remain NPCs in the postnatal stage and beyond [16]. Members of the bHLH family have emerged as important regulators of neural cell fate specification and differentiation. Additionally, Notch, Bone morphogenetic protein (BMP), Fibroblast growth factor (FGF), and Wnt signaling pathways regulate the expressions of bHLH factors.

Neurodegenerative diseases, including Huntington’s disease (HD), Parkinson’s disease (PD), and Alzheimer’s disease (AD), still have no cure. All these diseases occur during neurodegeneration and neuron death. The loss of neuronal tissues eventually results in severe effects on cognitive ability, motor control, and everyday function. Lately, many neuronal lineage conversion strategies for neurodegenerative diseases are using one or more of these *bHLH* genes. Most of the bHLH factors are involved in the conversion of resident astrocytes into functional neurons. Recent studies suggest adeno-associated virus (AAV)-mediated gene delivery for converting resident astrocytes to functional neurons [17,18,19,20,21,22,23,24,25]. However, stringent lineage-tracing methods have proven that the cell source of converted neurons is endogenous neurons, not resident astrocytes [26].

This review focuses on the role of bHLH transcription factors in the development of telencephalons in the CNS, maintenance of NPCs, and application of therapeutic strategies.

## 2. bHLH Transcription Factors Induce Neuronal and Glial Differentiation

The bHLH transcription factors, a large superfamily of transcriptional regulators, function in critical developmental processes such as the development of the nervous system. Members of the bHLH superfamily have two highly conserved and functionally distinct domains, which together make up a region of approximately 60 amino acid residues. The HLH domain consists of two α-helices connected by a non-conserved loop region. It is mainly involved in dimerization, but the basic domain directs DNA binding (Figure 1) [2]. The end of each amino-terminal region has the basic domain that binds the transcription factor to DNA at a consensus hexanucleotide sequence, also known as the E-box consensus sequences. The end of each carboxy-terminal region has the HLH domain that facilitates interactions with other protein subunits to form homo- and hetero-dimeric complexes. After dimerization, the bHLH transcription factors bind E-box motifs with the consensus sequences CANNTG, two central nucleotides, and surrounding nucleotides. It also provides specificity of binding [2]. The heterogeneity of the E-box sequence and the formation of dimers by different bHLH proteins determines the ability to control diverse developmental functions through transcriptional regulation. Based on their expression patterns, the bHLH transcriptional factors are classified into classes I and II [27,28]. The class II bHLH factors have tissue-specific expression profiles and are involved in various developmental processes [2].

We focus on the class II bHLH transcription factors that are expressed and involved in the development of the nervous system. Neural-specific *bHLH* genes are subdivided based on their homology to *Drosophila* genes, including the *achaete-scute complex* (*AS-C*) (*achaete, scute, lethal of scute*) and *atonal* (*atonal, amos, and cato*) gene families [2,29,30,31]. Single murine *achaete-scute-like 1* (*Ascl1/Mash1*) gene and the more distantly related *Nscl* family genes (*Nhlh1/Nscl1, Nhlh2/Nscl2*) are expressed in the nervous system. Multiple atonal-related *bHLH* genes that are expressed in neural lineages are the following: *Neurogenin* (*Neurog1, Neurog2, Neurog3*), *Neurod* (*Neurod1, Neurod2/Ndrf, Neurod6/Math2, Neurod4/Math3*), *Atonal* (*Atoh1/Math1, Atoh7/Math5*), and *Olig* (*Olig1, Olig2, Olig3, Bhlhe22/Bhlhb5*) members [32]. Not only is this categorization based on sequence similarities, but genes can also be categorized based on functional properties into proneural, neural and glial differentiation genes.

### 2.1. Function of the bHLH Gene in the Dorsal Telencephalon

It seems that Neurog1 and Neurog2 are expressed in cortical progenitors. Their expression is stronger in the Ventricular Zone (VZ) than the Subventricular Zone (SVZ) during the onset of neurogenesis in the dorsal telencephalon (Figure 2). During this period, Neurog2 continues to be expressed till after neurogenesis (E10.5~E17), while Neurog1 expression begins to decline in the later stage (after E15.5) [33,34]. Neurog1 transcripts are enriched in the lateral domains, whereas Neurog2 transcripts are prevalent in the VZ. It was revealed through loss-of-function experiments that Neurog2 generates a subset of the earliest-born (E10.5~E12.5) Cajal-Retzius neurons [33]. Interestingly, fewer Cajal-Retzius neurons were generated in Neurog2−/− cortices, whereas more early-born (E12.5~E14.5) neurons were generated in Neurog1−/− cortices [33,35,36]. The inhibition of the proneural activity of Neurog2 by Neurog1 indicates that in the absence of Neurog1, the proneural activity of Neurog2 increases and promotes the differentiation of more Cajal-Retzius neurons [33]. However, both genes are required together for generating Cajal-Retzius neurons in other regions, such as the lateral piriform cortex [33]. The subplate neurons are the next cells differentiated in the neocortex. The absence of Neurog2 function will result in decreased cell numbers and tissue collapse [37]. Interestingly, Neurog2 and Ascl1 form a genetic switch, and when Neurog2 is turned off, Ascl1 is turned on [34,35]. Because of this, the subplate is interrupted in Neurog2−/− cortices by upregulated expression of Ascl1, resulting in the misspecification of early-born neurons to an abnormal identity of GABAergic neurons [34,35]. Ascl1 overexpression leads to the induction of ectopic GABAergic interneurons and OPCs [34,35,38].

During the embryonic neocortex development before E14.5, most Nscl1 and NeuroD family genes are expressed in glutamatergic neuron subsets, and Neurog2 promotes their expressions in cortical and subcortical progenitors [36]. Thus, Neurog1/2 is essential for the specifications of glutamatergic neurons during telencephalon development. In the early cortical progenitors, glial fates are restricted by Neurog1, which promotes neurogenesis in two ways. First, the CBP-Smad1-Stat3 transcriptional complex is separated from the regulatory site of genes involved in astrogliogenesis by the Neurog1 protein [39]. Second, Neurog1 stimulates the production of miR-9, a microRNA targeting the Jak-Stat pathway components [40]. So far, Neurog2 has not been proven to have a function in the inhibition of gliogenesis, despite its structural similarity to Neurog1.

Ascl1 is limitedly expressed in cortical progenitors in the VZ and SVZ and is strongly expressed in the ventral telencephalon predominantly by NPCs [16]. GABAergic inhibitory neurons are known to be produced in the germinal zone and move tangentially to the neocortex [41,42]. GABAergic neuron formation is greatly reduced in Ascl1 knockout mice, but overexpression of Ascl1 causes ectopic GABAergic neuron generation from dorsal cortical NPCs [43,44,45]. Compared to Neurog2, Ascl1 is essential for the development of early-born glutamatergic Cajal-Retzius neuron subpopulations in the developing neocortex [46]. Ascl1 also regulates laminar fate determination through input into the cortical derepression circuit in the absence of Neurog2 [47]. To turn off the production of Tbr1 (a layer VI marker) in upper-layer neurons without the expression of this gene, Neurog2 and Ascl1 must work together. Ctip2+ layer V neurons are generated in Neurog2+/+ and Ascl1+/+ cortices, but they are lost in Neurog2−/− and Ascl1−/− cortices. Thus, sufficient expression of Neurog2 and Ascl1 is necessary [46]. Both the proneural genes, for example, Neurog2 and Ascl1, are required to regulate the timing of differentiation of neural cells. Moreover, Ascl1 is implicated in the determination of an OPC fate in cortical progenitors at the early postnatal stages [48].

### 2.2. Function of bHLH Gene in the Ventral Telencephalon

The ventral telencephalon is divided into the three progenitor domains—lateral (LGE), medial (MGE), and caudal (CGE) ganglionic eminences—at early embryonic stages (~E12.5) (Figure 2). All three domains produce unique populations of GABAergic neurons [49]. These progenitor domains generate the remaining projection neurons in the ventral telencephalon, and the projection neurons constitute the globus pallidus, amygdala and septum (MGE, CGE), and striatum (LGE, CGE) [50]. Additionally, ventrally derived neurons migrate to the dorsal region of the telencephalon. The MGE and CGE give rise to the populations of neocortical and hippocampal GABAergic interneurons, and the LGE gives rise to the population of interneurons of the olfactory bulb. *Ascl1* is the only proneural *bHLH* gene that is expressed in the ventral telencephalon. The transcripts of *Acsl1* are found in subcortical progenitors in the VZ and SVZ of the LGE, MGE, and CGE. For the development of GABAergic neurons, *Ascl1* is specifically essential in the MGE [41,42]. The smaller MGE in *Ascl1−/−* brains is attributable to a decrease in proliferation as well as reduced expressions of *Dll1* and downstream Notch effectors such as *Hes5*, showing the important link between proneural genes and Notch signaling [41,42]. *Ascl1* also determines the OPC fate in the embryonic ventral telencephalon [38]. To preferentially induce an oligodendrocyte fate, *Ascl1* inhibits the development of Dlx+ progenitors and the GABAergic neuronal fate [49]. The function of *Ascl1* promotes the proliferation of subpallial progenitors and is related to genes that are positive cell cycle regulators [51]. In general, proneural genes induce cell cycle exit and neuronal differentiation, but misexpression of *Ascl1* induces proliferation in certain cellular conditions [51,52]. According to a recent study, misexpression of *Neurog2* may induce proliferation in the neocortex [53]. Like GABAergic interneurons, OPCs are induced in the ventral telencephalon and migrate into the dorsal telencephalon during embryogenesis [50]. In the ventral telencephalon, *Olig1* and *Olig2* are expressed in OPCs and sustained during migration [54,55,56]. Additionally, the fate specification of oligodendrocytes needs *Olig1/2* in the ventral telencephalon [57].

## 3. Functions of bHLH Transcription Factors in NPCs during Telencephalon Development

### 3.1. The Function of the bHLH Gene in the Ventral Telencephalon

*Olig* genes are members of the bHLH family of transcription factors. *Olig* genes consist of *Olig1*, *Olig2*, and *Olig3*, which are expressed in both the developing and mature central nervous systems. Olig1/2 are identified as important factors in the fate choice of oligodendrocytes [58] and strictly regulate their differentiation, maturation, and myelination [32]. Extensive studies have established the functional roles of *Olig1* and *Olig2* in directing neuronal and glial formation during different developmental stages. *Olig3* is also characterized as a neuronal differentiation gene. However, it controls the differentiation of precerebellar neurons, brainstem nuclei, and dorsal spinal cord neurons [32,59,60,61]. OPCs, a subtype of glia and precursors to oligodendrocytes [62], were found to express the bHLH transcription factor family of *Olig* genes [58].

*Olig2* single and *Olig1/2* double knockout mice lack the oligodendrocyte lineage cells [57,63,64]. Contrastingly, the forced expression of *Olig1/2* in NPCs is enough to produce the OPCs [65,66]. *Olig1/2* has common functions in oligodendrogenesis, but *Olig2* plays a stronger role in choosing the pMN domain. Since *Olig2* is essential for the specification of oligodendrocytes in the developing spinal cord and hindbrain, the absence of *Olig2* causes the pMN domain to diminish [57,63,64]. The maturation of oligodendrocytes is affected when *Olig1* is absent [63,67]. The functions of *Olig2* in the production of oligodendrocytes in the developing telencephalon have been thoroughly analyzed and studied at different time points, from NPCs to OPCs and mature oligodendrocytes [68]. Like the non-overlapping roles of *Olig1* and *Olig2* in the spinal cord and hindbrain, there are no common roles for *Olig1* and *Olig2* in the proliferation and differentiation of NPCs and OPCs. While *Olig1* promotes the differentiation of OPCs, *Olig2* promotes the determination of OPCs and other types of neurons in the telencephalon [69,70].

The post-transcriptional modification of phosphorylation regulates the role of Olig2 during differentiation [58,71]. The activation to generate OPCs is caused by serine (S147) phosphorylation at the ST box of mouse Olig2 protein by CK2 kinase [72]. Serine phosphorylation in the bHLH domain plays a crucial role in motor neuron formation; the dephosphorylation causes the transition from motor neuron to oligodendrocyte formation in the pMN domain [73]. The phosphorylation in the bHLH domain stimulates the homodimer formation of the Olig2 protein. In contrast, dephosphorylation stimulates the heterodimer formation with the Neuro2 protein. Olig2 opposes differentiation but promotes the self-renewal of NPCs at early developmental stages. This self-renewing function of Olig2 is specifically regulated by developmentally controlled phosphorylation of a triple serine motif at the N-terminal region. Thus, depending on the phosphorylated status, Olig2 has contradictory functions: NPC proliferation and oligodendrocyte formation.

### 3.2. The Roles of bHLH in the Maintenance of NPCs

Neurons and glial cells are critical for functional CNS development. They regulate neural specification, synaptogenesis, synapse function, and circuit function. Glial cells primarily give support to neurons by providing nutrients to neurons. An appropriate proliferation of NPCs and intermediate progenitors is essential before differentiation [74,75]. Furthermore, brain processes such as learning and memory require several NPCs to be maintained in the hippocampus and lateral SVZ until adult brain development [16]. Hence, the NPCs remaining in these regions are essential for the development of brain functions and morphogenesis. In the undifferentiated state, NPCs are maintained by bHLH transcription factors, such as members of the Hes, Hey, and Id families (Figure 3) [16]. NPCs broadly express *Hes1* and *Hes5* in the VZ during telencephalon development, whereas *Hes3* is expressed during the early developmental state of the nervous system. The expressions of proneural genes, *Ascl1* and *Neurog2*, are directly repressed by Hes factors, and the absence of the *Hes* genes upregulates the expressions of proneural genes for neurogenesis promotion [76,77]. Hes factors also act as an antagonist of the proneural bHLH proteins by physical interaction. Neuronal target genes are repressed after the formation of the Hes-proneural bHLH complex, resulting in the repression of neurogenesis [78]. The expression of proneural factors is increased in *Hes1*/*Hes5* double knockout (KO) mice, resulting in severe premature neuronal differentiation, fast depletion of NPCs, and disordered nervous system patterns [77,79]. Despite the absence of *Hes1*, *Hes3* and *Hes5*, NPCs remain and proliferate almost normally in the developing telencephalon [76]. This demonstrates that different Hes factors are required in the telencephalon and other regions. Additionally, *Hey1* is highly expressed in the absence of *Hes1*, *Hes3* and *Hes5*, indicating that *Hey1* compensates for *Hes* to control telencephalon development. The Hes factors are known to increase the number of undifferentiated NPCs and impede neurogenesis [16].

The bHLH transcription activators, Ascl1, Neurog1, and Neurog2, activate neuronal differentiation in the developing nervous systems [2,80]. The expressions of ligands for Notch signaling, such as transmembrane proteins Delta-like (DII1) and Jagged1 (Jag1), are upregulated by these factors [81,82,83]. Before the activation of Notch signaling, the transmembrane portion and nucleus form complexes with the DNA binding protein RBPj κ and the co-activator Mastermind-like (Maml), releasing the Notch intracellular domain (NICD) [84]. NICD-RBPj κ-MAML is a transcriptional activator that drives the expression of bHLH transcriptional repressors, such as Hes1 and Hes5, and generates the canonical pathway. Conversely, the expression of Hes3 generates a non-canonical pathway of Notch signaling [76]. Hes factors, repressing the expression of proneural genes and *DIl1*, control neuronal differentiation and promote the maintenance of NPCs. Therefore, differentiating neurons suppress the neighboring cells from differentiating into the same cell type. This process, also known as lateral inhibition, prevents NPCs from simultaneous differentiation [85].

In addition to Hes factors, NPCs in the developing brain express Id factors. NPCs exit the cell cycle early and undergo rapid neuronal differentiation in Id1/Id3 double KO mice [86]. As a result, Id factors limit the premature differentiation of NPCs into neurons. Conversely, overexpression of Ids in NPC culture inhibits neurogenesis, suggesting that Id factors are adequate for NPC maintenance [87].

## 4. Functions of bHLH Transcription Factors in Reprogramming

### 4.1. Overview of Neuronal Reprogramming

Research in developmental biology has paved the road for direct neural cell reprogramming by identifying intrinsic and extrinsic cues that induce neurogenesis and gliogenesis [88]. Improved neural cell reprogramming techniques have facilitated their use for treating neurodegenerative disorders. There has been an increased focus on using *bHLH* genes, especially proneural transcription factors, for reprogramming cells.

### 4.2. Fibroblasts

Stepwise lineage conversion strategies have been developed for reprogramming somatic cells into induced neuronal (iN) identities. Neuronal reprogramming is mediated by bHLH transcription factors. In P19 cells, Ascl1, Atoh1, Neurod1, Neurod2, Neurod4, Neurod6, Neurog1, Neurog2, and Nhlh1 were shown to induce neurogenesis [89]. Initial reports of direct neuronal reprogramming in fibroblasts used Ascl1, Brn2, and Mytl1 (BAM) [90]. As the field of neuronal reprogramming expanded, Ascl1 emerged as a significant reprogramming factor to convert mouse and human fibroblasts [90,91,92,93,94,95], hepatocytes [96], cardiomyocytes [97], or astrocytes [98] to neurons.

When fibroblasts are transfected with *Ascl1*, they generate iNs [99]. Contrastingly, when *Ascl1* connects with *Pitx3*, iNs become dopaminergic [90]. Additionally, the overexpression of *Ascl1* with *Neurog1*, *Brn2*, *Myt1l*, *Isl1*, and *Klf7* leads to nociceptor-like sensory neurons [100]. Like *Ascl1*, *Neurog2* also has the ability to reprogram fibroblasts when co-transfected with *Sox11*, *Lhx3*, and *Isl1* [101]. Conversion to neural lineages using bHLH transcription factors is not exclusive to neurons. *Olig2* and *Sox10* with either *Zfp536* or *Nkx6.2* have also been used effectively to create OPCs from fibroblasts [102,103].

### 4.3. Embryonic Stem Cells and Induced-Pluripotent Stem Cells

Neural induction is initiated in pluripotent cells in a stepwise manner: first, BMP and activin/TGFβ signaling is blocked to produce induced neural stem cells (iNSCs), after which neuronal differentiation is initiated by adding various small molecules and signaling molecules [104]. Techniques producing striatal [105], dopaminergic, and spinal cord-like neurons [106] have been created based on signaling molecules that can produce neurons with certain phenotypes.

Overexpression of bHLH transcription factors is used to increase the efficiency in neuronal differentiation; for example, overexpression of Neurog1 and Neurog2 in human induced pluripotent stem cells (hiPSCs) leads to induction of neuronal differentiation [107]. Also, the combination of Neurog1 with Islet1 and Lhx3 in human embryonic stem cells (ESCs) converts the ES cells into motor neurons [108]. Lastly, Neurog2 alone can convert human ES or iPS cells into neurons [109].

### 4.4. Glial Cells

Several bHLH transcription factors have proven effective in transforming astrocytes into neurons. Astrocytes can develop into neurons with the help of Neurog2, Ascl1, and NeuroD1 [43,110,111]. The expression of *Olig2* inhibits the glial cells from transforming into neurons, but the knockdown of *Olig2* using *miR*-*Olig2* can reprogram glial cells into neurons [112]. Ascl1 transforms astrocytes into neurons with glutamatergic and GABAergic identities [18] when transfected into astrocytes of the dorsal midbrain. Contrarily, instead of inducing neurogenesis, Ascl1 promotes oligodendrogliogenesis when it is overexpressed in the adult neocortex, hippocampus, and spinal cord [113,114].

## 5. Signaling Pathways Related to Regulation of bHLH Transcription Factors

### 5.1. Notch Signaling

bHLH transcription factors, particularly those with proneural functions such as Ascl1, Neurog1, and Neurog2, have been associated with the Notch signaling system (Figure 4A). These factors also increase the ligand expressions of Notch, such as Dll1 and Jag1, to repress the proneural expression through lateral inhibition by binding Notch receptors on neighboring cells [81]. Binding of the Dll1/3 ligand results in the cleavage of the Notch receptor; the NICD region is then translocated into the nucleus, where it forms a complex with the coactivator Maml and the DNA-binding protein RBPjk [74,75,115]. The complex acts as transcriptional repressors, such as Hes1 and Hes5, via the canonical pathway. A noncanonical pathway induces the expression of Hes3 [76]. Hes factors inhibit the expression of proneural genes and Dll1, resulting in the suppression of neuronal differentiation and NPC maintenance [16]. This lateral inhibition prevents all NPCs from simultaneous differentiation, allowing NPC maintenance during the later phases of development.

### 5.2. Bone Morphogenetic Protein (BMP) Signaling

The bone morphogenetic protein (BMP) signaling pathway is important in the development of the nervous system. BMP signaling is involved in the creation of the neural crest and induction of both neuronal and glial fates from NSCs or NPCs in the cortex, hippocampus, midbrain, hindbrain, and spinal cord (Figure 4B). However, restriction of the BMP signaling is required to form the neural plate [77]. BMP4 signaling is especially important during early embryonic development in mice [78]. BMPs regulate both neuronal and glial development in embryonic NPCs in a time-dependent manner. The SMAD-independent MAPK/extracellular signal-related kinase (ERK) pathway is related to the neurogenic activity of BMP4 during forebrain cortical neurogenesis. BMP signaling also inhibits oligodendrogliogenesis by acting on Olig1/2 proteins in an indirect manner. The neurogenic action of BMP4 reduces the neurogenesis progression, and its dual role as an astrogliogenesis promoter and an inhibitor of oligodendrogliogenesis becomes increasingly apparent [79]. Surprisingly, the degradation of Ascl1 is also induced by BMP2, and the expression of Id is regulated by BMP signaling but not Notch signaling.

### 5.3. Wnt Signaling

As a major regulator, Wnt signaling plays an important role in the regulation of the transcription of *bHLH* genes (Figure 4C). Besides, many cellular processes, controlling cell proliferation, differentiation and migration, neural patterning and organogenesis, are also regulated by Wnt proteins during embryonic development [86]. Wnt signaling stimulates the *Neurog2* expression and is a direct regulator of *Atoh1* and *Neurog1* [116,117,118,119]. Ectopic expression of *Neurog1/2* is induced by components of Wnt signaling in the ventral telencephalon [116,117,119,120]. Wnt signaling regulates the *Neurog2* expression only during the early cortical neurogenesis, thereby temporarily restricting the regulation of *Neurog2* by Wnt [117,121]. Wnt signaling induces hippocampal neurogenesis by regulating *NeuroD1* expression and increases *Ascl1* expression in cultured cells [122,123,124].

### 5.4. Fibroblast Growth Factor (FGF) Signaling

Fibroblast growth factors (FGF) serve many functions in both developing and adult organisms (Figure 4D). Particularly, humans contain 18 FGF ligands and four FGF receptors, also known as *FGFR* [125]. FGF is a group of 22 polypeptides with biochemical properties that are structurally and physiologically important [126]. They regulate several developmental processes, including cellular proliferation, differentiation and migration, morphogenesis, and patterning. The mammalian FGFR family consists of four members: FGFR1, FGFR2, FGFR3, and FGFR4. These tyrosine kinase receptors FGFR1, FGFR2, FGFR3, and FGFR4 are used by FGF ligands to trigger their activities [127]. The signals are transduced by under activation of the FGFRs in three major pathways: RAS/MAPK, PI3k/AKT, and PLC [128,129]. FGF signaling activates RAS/RAF/MEK/ERK signaling, which leads to the repression of *Neurog2* expression and the activation of *Ascl1* expression [52]. *Neurog2* promotes the differentiation of glutamatergic neurons in the cortex, while ectopic *Ascl1* determines the fate of GABAergic neurons or OPCs [52].

## 6. Application of bHLH Transcription Factors in Therapeutic Strategies for Neurodegenerative Diseases

As reviewed above, various types of bHLH transcriptional factors regulate neuronal or glial cell fate. There is no ability to generate neurons in the majority of adult brain regions except within neurogenic niches [16]. Most neurodegenerative diseases, such as HD, AD, PD, and ALS, have characteristics of the progressive formation of inclusion bodies and the loss of specific neuronal populations, structure, and function, which eventually result in death. New therapeutic strategies, such as cell or gene therapy, are the fastest-growing techniques using bHLH transcription factors.

An improvement in the repair response of defective and damaged tissues using stem cells is called stem cell therapy (regenerative therapy) [130,131,132,133]. Recent studies have reported the ability of NPCs to differentiate into limited cell lineages, such as neurons, astrocytes, and oligodendrocytes [16]. Currently, many studies are focusing on investigating the therapeutic effects of neurons, astrocytes, and oligodendrocytes that are differentiated from NPCs [131] and then develop through spontaneous transdifferentiation in the brain of patients [134,135]. Additionally, several studies have established the effect of mesenchymal stem cells (MSCs)-derived neurons and the promotion of NPCs by secretion of neuroprotective factors from MSCs [136,137,138,139]. The new therapeutic strategies apply stem cell technology to provide de novo synthesis and the delivery of neuroprotective growth factors, such as glial cell line-derived neurotrophic factors (GDNF), brain-derived neurotrophic factors (BDNF), insulin-like growth factor 1 (IGF-1), and vascular endothelial growth factor (VEGF) [140]. According to recent studies, most stem cell-based therapies for neurodegenerative diseases are based on preclinical experiments from animal models. Researches show how stem cells influence endogenous cells, differentiate into neuronal and glial cells, and improve motor and memory impairments [131,132,141]. Besides, many results have shown how stem cell therapies can induce neurogenesis in patients with neurodegenerative disorders. However, since most of the differentiation in functional neurons is spontaneous in vivo, there is a need to understand the mechanisms of each neurodegenerative disease. For example, it is necessary to understand what type of neurons are lost, where the loss of neurons occurs, and how exactly neurogenesis works. If we focus on different types of neurons required for neurodegenerative diseases by utilizing bHLH transcription factors, this therapy has a huge potential to become a more advanced and developed strategy.

Gene therapies also have therapeutic potential directed toward the correction of pathogenic mechanisms, neuroprotection, and neurogenesis in neurodegenerative diseases. Therefore, it is important to understand the knowledge of disease pathogenesis and gene expression of spatial and temporal specificity in therapeutic efficacy. Generally, bHLH transcription factors, such as Dlx1, NeuroD1, and Neurog2, are delivered through stereotaxic injections into specific brain regions via AAV [17,18,19,20,21,23,24,25,142,143]. Recently, new technologies such as MRI (Magnetic Resonance Imaging)-guide convection-enhanced delivery (iMRI-CED) have been applied for more precise surgery [140]. bHLH transcription factors such as Dlx1, NeuroD1, and Neurog2 are mainly used for the conversion or transdifferentiation of resident astrocytes into functional neurons. For the specific targeting of reactive astrocytes, a method for transdifferentiation of only a specific cell type into functional neurons through a system called the TRANs-Cre-DIONE (Transdifferentiation of Reactive Astrocytes to Neurons by split-are using *GFAP* and *Lcn2* turning on of DIO-Neurogenin2 under *Ef1a* promotor) is also studied [144]. At the preclinical level, the in vivo conversion and transdifferentiation experiments using AAV vectors confirmed the function of converted and transdifferentiated neurons and the improvement of the behavioral deficit in the mouse disease model [24,25]. However, according to recent findings, reporter-positive neurons generated through AAV-mediated AtN conversion do not originate from resident astrocytes but from endogenous neurons. It was found that reporter expression occurred in endogenous neurons through methods such as stringic lineage tracing [26]. The answer explaining the improvement of the behavioral deficit is said to be due to a neuroprotective mechanism [26].

However, since the mechanism remains unclear, additional research is needed. Gene therapy can be a more powerful therapeutic approach, provided only specific types of cells can be targeted and converted into functional neurons when problems such as leaky expression are solved.

## 7. The Challenges of Cell and Gene Therapy and Future Directions

### 7.1. Cell Therapy

As stem cell-based therapies relate to the CNS, they provide many opportunities against intractable diseases and injuries. Cell therapies for different diseases are slowly but steadily progressing and have already reached or are about to reach clinical trials. However, stem cell-based therapies face hurdles and the potential risk of oncogenesis, heterogeneity, and immune rejection [145,146]. One of the most important benefits of stem cells is their infinite proliferation, but this property is a doubtful advantage. If the cells keep proliferating after transplantation, this may ultimately result in the formation of tumors [133].

Each stem cell line is unique. An important element in stem cell therapy is decreasing the probability of heterogeneity. To successfully decrease the chance of heterogeneity, it is important to use stem cells from single-cell colonies. Each cell line has different morphology, growth curve, gene expression, and tendency to differentiate into different cell lineages [133]. Currently, certain factors have been identified that can reduce heterogeneity, such as bHLH transcription factors. But, in order to fully cover heterogeneity in stem cells, more such factors are required.

Additionally, immune rejection is also a serious challenge in stem cell therapy. To minimize the risk of immune rejection, Japan and other countries are working to create and update the banking of iPSCs from donors with homozygous human leukocyte antigens (HLA) [147]. Immune rejection in cell therapy causes the destruction of the transplanted cell, which may result in cell death; for example, they suppress immune cell activation through the genetic transduction of co-inhibitory molecules or inducing immune regulatory cells [147].

Gene therapy plays an essential role in order to overcome the challenges of cell therapy, such as oncogenesis, heterogeneity, and immune rejection.

### 7.2. Gene Therapy

Neurodegenerative diseases, such as HD, AD, PD, and ALS, cause neuronal death due to different causes. Gene editing using CRISPR-Cas9, gene silencing using small interference RNA (siRNA) and antisense oligonucleotides (ASOs), and in vivo AAV gene therapy using bHLH transcription factors packaged in AAV vector as cell fate conversion is still being developed for gene therapy.

However, each approach presents its own challenges. First, CRISPR/Cas9 has off-target effects (OTEs) for editing other sequences rather than the desired sequences [148]. Also, because the expression of synthetic siRNA is temporary, siRNA is loaded into a viral vector such as AAV to solve the problem. In fact, CRISPR/Cas9 and shRNA are being applied in vivo for astrocyte-to-neuron conversion (AtN) studies that convert striatal astrocytes into functional neurons through the knockdown of the *PTBP1* gene [25].

As a recently emerging novel therapeutic approach, various AtN conversion studies are being conducted. Using an AAV vector, bHLH transcription factors such as *NeuroD1*, *Dlx2*, and *Neurog2* are specifically expressed in astrocytes and converted into functional neurons. As a result, histological, electrophysiological, and behavioral improvements in the mouse disease model have shown potential as a new treatment for neurodegenerative diseases [23,24,142,144].

However, using diverse multiple stringent lineage-tracing methods, the reporter was expressed in the endogenous neurons but not on the newly converted neuron originating from the resident astrocyte [26]. In order to target astrocytes more specifically, a method such as TRANsCre-DIONE has been studied [144], but validation is necessary because it has not been verified with the previously verified lineage-tracing methods.

It is expected that the behavioral improvements in the mouse disease model shown in the previous studies may be influenced by neuronal protection. In the stringent lineage-tracing methods, leaky expression of the reporter gene was also observed in Cre-inducible transgenic mice expressing only certain astrocyte promoters such as *Aldh1l1* and *Gfap* [26].

Although the cause of reporter expression in endogenous neurons is not yet clear, we predict that the cause is tunneling nanotubes (TNTs). TNTs are a mechanism used for intercellular communication between astrocytes and neurons [149]. Through TNTs, not only pathological proteins such as mutant Huntington protein (mHTT) but also substances such as GFP and microRNA can be moved; hence, the possibility of moving Cre proteins with similar sizes can be considered [149]. In fact, Cre protein was made specifically only in astrocytes, but it is necessary to check the possibility that Cre protein was displaced and non-specific expression was formed by creating connections with the surrounding endogenous neurons through TNTs.

In the future, cell fate conversion studies, such as in vivo AtN conversion, must be accompanied by lineage-tracing methods. Although time-lapse imaging [150] is the best way, it is difficult to establish in most laboratories due to technical and cost limitations [26]. Since the newly converted neurons go through an immature stage, these neurons need to be characterized using Doublecortin (DCX)—the marker of immature neurons [151]. Therefore, wide use of common methods such as DCX, BrdU-tracing method, and single-cell RNA sequencing (scRNA-seq) should be examined for in vivo cell fate conversion studies [26].

## Figures and Tables

**Figure 1 ijms-23-13936-f001:**
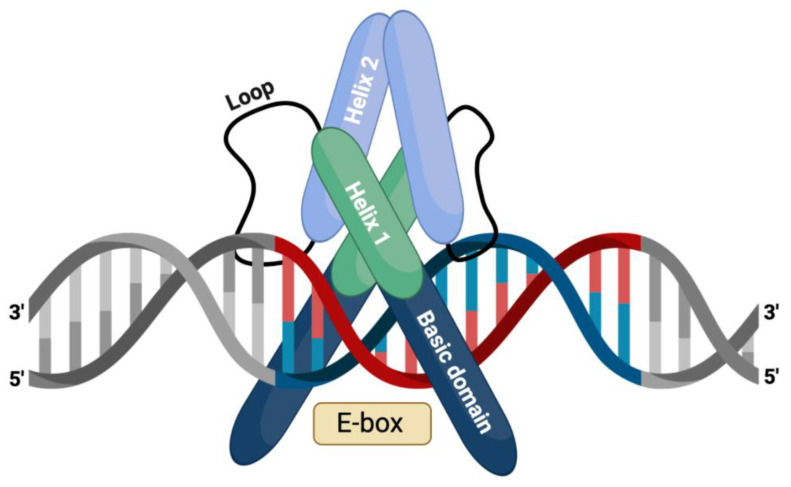
The structure of the bHLH transcription factor complex involved in binding to the E-box sequences of the DNA.

**Figure 2 ijms-23-13936-f002:**
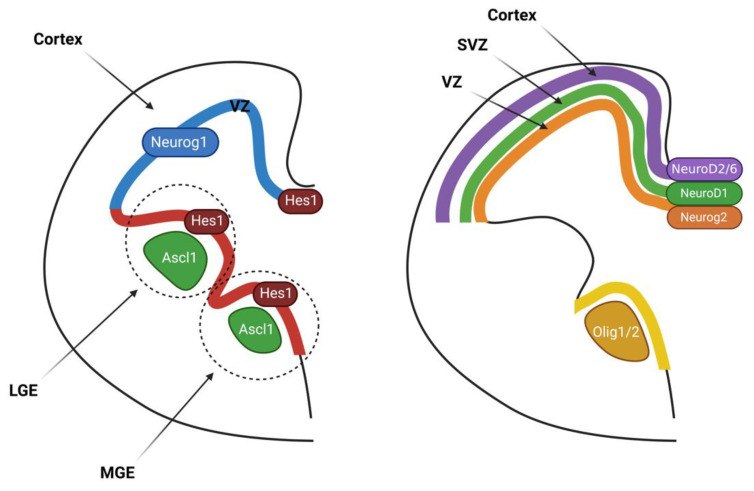
The rodent ventral and dorsal telencephalon. The left side of the figure is the rodent ventral telencephalon. The right side of the figure is the rodent dorsal telencephalon. In the ventral telencephalon, the *Ascl1* gene (LGE, MGE) is expressed. Glial progenitors and oligodendrocyte precursor cells (OPCs) induce the expression of *Olig1/2*. In the dorsal telencephalon, the cortical progenitors induce the development of the neocortex and the expression of *Neurog1* and *Neurog2* (VZ). The mature neurons induce the expression of *NeuroD* genes, such as *NeuroD1* (SVZ), *NeuroD2*, and *NeuroD6* (cortex).

**Figure 3 ijms-23-13936-f003:**
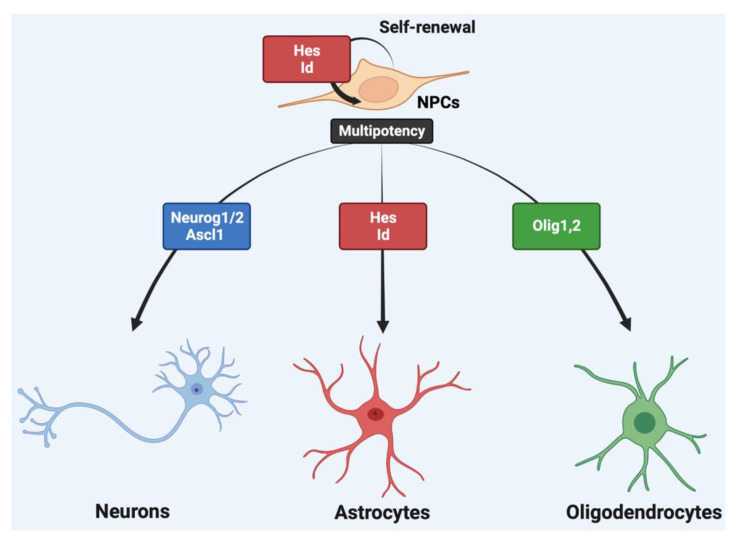
The choice of cell fate and self-renewal of NPCs is regulated by bHLH transcription factors.

**Figure 4 ijms-23-13936-f004:**
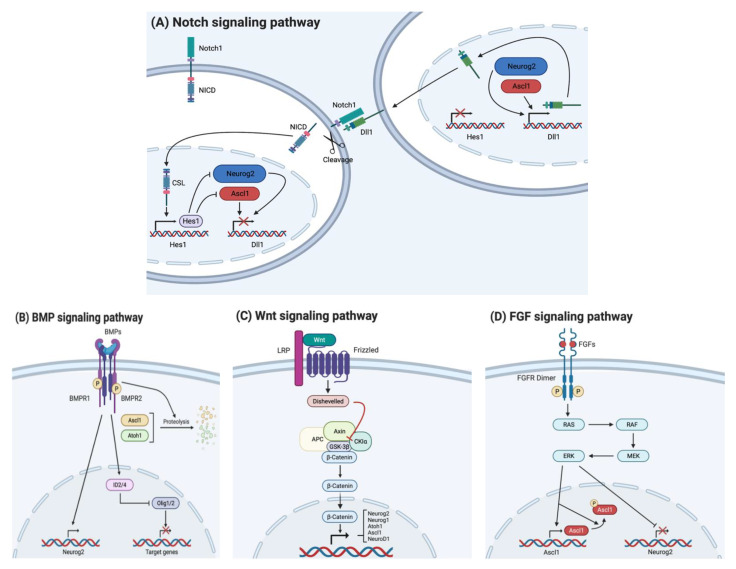
bHLH transcription factors are regulated by a variety of signaling pathways: Notch, BMP, Wnt, and FGF signaling. (**A**) Proneural transcription factors induce the expression Dll1 ligand in the emitting cell (**right**). Dll1 ligand binds to the Notch1 receptor on the receiving cell (**left**), which causes the NICD to be cleaved and translocated to the nucleus. After translocation to the nucleus, NICD interacts with CSL to initiate the transcription of *Hes1*, which inhibits the expression of proneural genes. (**B**) BMP ligand binds to serine-threonine kinase receptors, BMP receptors, which initiate the expression of *Neurog2* and repress the expression of *Oilg1/2* through activation of ID2/4. Simultaneously, Ascl1 and Atoh1 are broken down by proteolysis. (**C**) Interaction of Wnt with the transmembrane receptor, frizzled (*FZ*) and low-density lipoprotein receptor (LRP) activates disheveled (DVL), which can induce the release of β-catenin from the β-catenin destruction complex (stabilization). After the stabilization of β-catenin, it translocates to the nucleus to initiate the expression of numerous *bHLH* genes. (**D**) FGF ligand binds to receptor tyrosine kinase (RTK) to induce dimerization which activates RAS, RAF, MEK, and ERK signaling. These signaling cascades induce the expression of *Ascl1* and repress the expression of *Neurog2*.

## Data Availability

Not applicable.

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
