# Peer review of "The bHLH Transcription Factors in Neural Development and Therapeutic Applications for Neurodegenerative Diseases"

_ijms, 2022, doi:10.3390/ijms232213936_

Round 1

Reviewer 1 Report

Manuscript ID ijms-1950824, Lee DG and Baek K-H

The bHLH Transcription Factors in Neural Development and Therapeutic Applications for Neurodegenerative Diseases

Treatment of neurodegenerative diseases such as Alzheimer’s disease (AD) includes some strategies of cell therapy employing stem cells and gene therapy through cell fate conversion. In this paper, the authors review how bHLH acts as a regulator in neuronal differentiation and reprogramming and the current strategies that regulate cell fate determination as well as the importance of bHLH factors to direct neuronal and glial cell fate specification and differentiation. The authors also investigated the limitations and future directions of conversion or transdifferentiation using the ability of bHLH factors.

Given the increased neurogenesis in neurodegenerative diseases, I agree that the role of bHLH factors might be important, and might be useful for therapy of these diseases. My questions (minor) are as follows. Please consider these comments before acceptance of the paper.

1.     Why bHLH is important compared to other transcription factors?

2.  Are there any types (developmental, early and late ) of neurodegenerative diseases that are associated with the polymorphism of bHLH genes?

3.     More explanation of induced pluripotent stem cells (iPSCs) might be required.

Reviewer 2 Report

In this work, the authors describe the potential therapeutic applications of bHLH transcription factors, describing their role during telencephalon development and corresponding signalling pathways. The main contribution of this work is related to the cell and gene therapy (Section 6-7) involving bHLH proteins. Current approaches with the most recent achievements (published in the last 1-2 years) are summarized. The first part of this review is more descriptive, explaining already known signalling pathways (Section 5) involved in the bHLH proteins’ activity and regulation.

However, the English language presents many issues throughout the manuscript, and it should be extensively rearranged. Figures are not cited in the text and many of them are not appropriately described in the figure legends (many abbreviations reported in the figure are missing in the description). Therefore, the manuscript should be revised significantly in order to be accepted for publication.

Comments and suggestions are listed below. 

Abstract

Line 17: the second part of the sentence is unclear; the current strategies “are focused on” (instead of “…strategies that play a critical role”)   

Line 21: remove “the ability of” (otherwise specify which kind of ability)

Introduction

Line 26 – add “family of” between helix-loop-helix and proteins

Line 27: “The helix-loop-helix (HLH) domain is directly followed by the N-terminal basic region in the bHLH transcription factor”-unclear sentence; HLH domain is situated at the N-terminus and is comprised of basic amino acid residues. This is the fundamental description of bHLH TFs and it should be clear.

Line 30: TFs is the abbreviation of transcription factors (replace “transacting” with the appropriate definition)

Line 39: regulators of neural cell fate (instead of regulators in neural cell fate)

Line 40: fibroblast growth factor is more often abbreviated into FGF (in some part of the work capital letters were used while here small print was used; for consistency, only one way should be used throughout the manuscript)

Line 41: remove “in” between Wnt and signalling pathways

Line 44-45: unclear sentences that should be rephrased since neurons can die following traumatic brain injury as well; neurodegeneration itself (before the cell death) can cause disease etc.

Line 52: “the cell source of converted neurons is endogenous neurons, not resident astrocytes” - unclear meaning 

Line 63: The fact that bHLH TFs are “found in organisms” is obvious – remove it or explain better

Line 80: “involved” (instead of “function”) in the development

Line 90: the distinction “proneural and neural (neuronal or glial)” is unclear-explain better (moreover, “neural” and “neuronal” are synonyms!)

Line 98 : Section 2.1 should cite Figure 2 in the text (as all other figures that are not mentioned anywhere). It is also not clear which model is used (human or rodents?)-this should be clearly stated for instance at the end of the first sentence of this paragraph as well in the title of the Figure 2.

Figure 2 has additional issues: LGE, MGE, VZ, SVZ are not explained it the figure legend; Olig1/2 are mentioned in the legend but in the figure only Olig 2 is reported. The description of left and right part of the figure is missing.

Line 99: as it is written, it seems that Neurog1 and Neurog 2 are cortical progenitors-this sentence should be corrected accordingly.

Line 104-108: it is not clear the difference between “earliest-born” and “early-born” neurons

Line 129-130: replace “…where Ascl is predominantly expressed by NPCs” with “… predominantly by NPCs” to avoid unnecessary repeating

Line 165-167: “in certain conditions” – avoid repeating; otherwise explain more precisely in which conditions

Line 182: the use of “novel” it’s confusing since Olig genes were identified more than 2 decades ago (see ref. no. 48) while the citing article (ref. no. 50) was published in 2012

Line 186-187: “Olig2 mostly plays a stronger role…” – mostly or stronger?

Olig 3 is not described at all.

Line 211-213: Section 3.2. should cite Figure 3. The first sentence is oversimplified – sufficient number of glia and neurons is critical for proper/functional/correct CNS development  

Line 260: plural form should be used, as for all other cells

Line 269: iNs (instead of iNS)

Line 278: iNSCs should be used (instead of NSCs)

Line 298: Section 5 should cite Figure 4 and its panels in the text, whenever is appropriate (Figure 4A for the subsection 5.1, Figure 4B for 5.2 etc). The figure legend should be corrected in order to be consistent: Fgf (replace with capital letters), genes are reported in italic font, but it is not clear why β-catenin and Olig1/2 (proteins) are written in the same way. This should be corrected for all other genes (italicized) and proteins (not italicized).

Line 338 and 339 : Fgf – replace with FGF (as it has been displayed in the Fig.4D)

Line 369: The sentence “Because of this, neurons are not able to naturally replenish after death.” should be rephrased (it’s because they are postmitotic cells; “naturally” is not specific enough).

Line 371: ALS was not previously defined in the manuscript (while HR, AD and PD were mentioned in the abstract); unclear part of the sentence “…have characteristics of progressive inclusion bodies” should be corrected, for example “…have characteristics of progressive formation of inclusion bodies”

“resulting to death” – this should be rephrased as well

Line 374: unclear meaning of “due to the characteristics of neurons” (which characteristic? this part should be removed or explained better)

Line 381-383: if only one specific work is cited, then it is not correct to claim that “several studies have established….”

Line 393: understanding of the mechanisms

Line 405: define MRI at the first mention

Line 408:  “the specific targeting of reactive astrocytes” or “the specific reactive astrocytes’ targeting”

Line 420-423: check grammar (past/future tense)

Line 429 and 449: oncogenesis and tumorigenesis are synonyms! Mutagenesis not necessarily leads to tumorigenesis but as it is written (“oncogenesis (mutagenesis and tumorigenesis)” it is misleading

Line 434: the statement “no two stem cells are identical” is strongly arguing against basic stem cell property, i.e. self-renewal (two identical cells generated by the progenitor cell); this sentence should be rephrased

Line 443: explain better the (questionable) statements in this sentence: „Immune rejection in cell therapy causes the destruction of the transplanted cell as a whole and leads to tumorigenesis.“ Tumorigenesis occurs as a consequence of intrinsic proliferative potential of stem cells so it's not clear the link with immune rejection.

Line 461: replace „Astrocyte to Neuron“ with „astrocytes-to-neuron“ (small letters)

Line 478-480: unclear sentences (e.g. “one of our prediction” – prediction of what?)

Line 492: “wide/wider use” instead of “more utilization”

Round 2

Reviewer 2 Report

Line 130 of the revised manuscript (and Response #15 provided by the authors): The new sentence „It seems that Neurog1 and Neurog2 are cortical progenitors“ is wrong: Neurog1 and Neurog2 are expressed in cortical progenitors. (They are not cortical progenitors, but genes encoding for bHLH transcription factors that are expressed in cortical progenitors). Consequently the next sentence should begin with „Their expression is stronger....“ instead of „They are greater...“

The revised manuscript is significantly improved.
